# The Effect of Sodium Bicarbonate Injection on the Physico-Chemical Quality of Post-Harvest Trout

**DOI:** 10.3390/foods12132437

**Published:** 2023-06-21

**Authors:** Koray Korkmaz

**Affiliations:** Fatsa Faculty of Marine Sciences, Department of Fisheries Engineering Technology, Ordu University, 52400 Ordu, Turkey; koraykorkmaz@odu.edu.tr; Tel.: +90-452-4235053 (ext. 4720)

**Keywords:** *Oncorhynchus mykiss*, sodium bicarbonate, post-harvest, quality, storage

## Abstract

The muscle hardness of fish is an important parameter associated with meat quality, and the post-mortem decrease in the pH of fish tissue pH affects its physical properties. We hypothesized that maintaining a high pH in fish tissue after death would prevent protein denaturation and consequent textural deterioration. This study aimed to determine the effectiveness of sodium bicarbonate (SBC) injections in preventing tissue softening caused by low pH after death in trout. We injected varying molar concentrations of SBC in rainbow trout (*Oncorhynchus mykiss*; 0 M, 0.5 M, 0.75 M, and 1 M) after harvest, and the product quality was assessed at 0, 24, 48, 72, and 96 h of ice storage. Quality was evaluated using proximate analyses for color, pH, water holding capacity (WHC), texture profile, and rigor index. The 0 M group had the lowest pH, and the 0.75 M group had the highest pH at all time points during storage. We observed improved tissue texture during storage in fish treated with 0.75 and 1 M SBC. The texture profile analysis showed higher hardness, frangibility, and stickiness in the tail than in the other regions. These varying results can be explained by significant differences between parts of the fish and sampling point selection. We also observed the highest pH and WHC values in the groups injected with 0.75 and 1 M SBC during storage.

## 1. Introduction

Muscle hardness is an important parameter for determining the quality of fish, affecting its acceptability for consumption [1]. Muscle hardness is mainly associated with myofibrillar muscles containing myosin and actin in the connective tissue consisting of collagen [2]. Softened fish muscle is considered unsuitable for consumption by consumers [3]. The texture is one of the main parameters involved in the physical assessment of fish quality by producers and consumers [4], and several studies have evaluated changes in fish tissue throughout the degradation process [5,6]. Recent studies have used instrumental texture or tissue profile analysis (TPA) as the evaluation method [7,8]. Several studies on the instrumental tissue analysis of salmonids have also been conducted [9,10]. As described by Rosenthal [11], a tissue analyzer can measure primary instrumental tissue properties (such as hardness and cohesiveness). However, the hardness, springiness, stickiness, flexibility, and chewiness parameters have recently emerged as the most widely accepted and studied properties in fish tissue assessment [5,12,13].

The post-mortem decrease in fish muscle pH affects the meat’s physical properties. As the pH decreases, the net surface load on the muscle proteins decreases, which partially denatures the proteins and reduces their water holding capacity (WHC). A decrease in the WHC of proteins has a significant effect on the texture of fish, and restoring the pH of meats has been shown to improve the WHC [14].

The WHC of muscular foods is important for several reasons and depends on heat-induced structural changes, sarcomere length, pH, ionic strength, osmotic pressure, and rigor mortis status [15]. The most prominent quality parameters for the consumer are flaking texture, tenderness, and juiciness. These parameters are all associated with the WHC because of the dependence on the heat denaturation of proteins. The WHC has also been used to measure quality [16]. Sodium bicarbonate (SBC) can solve various challenges, such as moisture loss and cooking loss, by increasing the WHC [17].

Rigor mortis is the most important biochemical phenomenon affecting the initial quality of fish after death. Immediately after the death of the fish (pre-rigor), fish muscle contains various chemicals that keep it elastic for several hours, such as phosphocreatine, adenosine triphosphate, and glycogen. Upon death, blood circulation and immune systems cease to function, resulting in an interruption in oxygen supply and the onset of anaerobic degradation of glycogen, known as glycolysis [18]. A post-mortem decrease in the pH of muscles significantly affects the physical properties of the fish meat [19]. These textural changes in fish muscles are critical to sensory acceptance for producers and consumers. Textural degradation, such as excessive softening after harvest, is undesirable as it prevents the fish from being processed by hand or machine. A low pH causes the fish’s meat to harden after death. However, a rapid decrease in pH after death causes meat softening.

SBC is a ‘generally recognized as safe’ food ingredient and has been widely used in nutrition and industrial processes for several years [20]. SBC contains a synergistic anion (HCO_3_–) and a strong buffer capacity and is widely used in meat products [21]. Using SBC in salted meat products increases their WHC, tenderness, flavor, and cooking efficiency by increasing the pH and ionic strength in products such as fillet chops and chicken breast meat [22,23]. SBC is mostly used as a cryoprotectant in minced seafood, particularly in surimi, to render the meat a white color and to increase the WHC. Lopkulkiaert and others [24] investigated the effect of SBC and trace amounts of citric acid on meat yield during frozen storage and cooking loss after thawing in shrimp. Their study reported reduced water loss caused by freezing, increased water absorption, increased yield after freezing and thawing, and decreased freezing loss. In addition, the hardness of the samples cooked after freezing reportedly decreased, and the WHC increased. Furthermore, Martinez-Alaverez and others [25] found that soaking cod in alkaline SBC solutions improved its protein-bound functional properties. In addition, they proposed that the industrial use of SBC would be remarkable as it yields significant juiciness, solubility, and tissue hardness.

To the best of our knowledge, very few studies have been conducted on the use of SBC in the post-harvest seafood industry. SBC is mostly used in poultry and pork products. The production and consumption of rainbow trout (*Oncorhynchus mykiss*) are increasing rapidly in Turkey. In this study, the changes in the physical, chemical, and proximate composition parameters of rainbow trout with regard to the effect of various concentrations of SBC were investigated post-harvest. We aimed to maintain a high pH value in fish tissue after death to prevent protein denaturation and consequent textural deterioration caused by a low pH.

## 2. Materials and Methods

### 2.1. SBC Applications

The rainbow trout were obtained from a commercial farm in the Ordu province of the Eastern Black Sea Region. In total, 100 sampled fish were studied. The mean length and weight of sampled fish were 32 ± 0.8 cm and 292.6 ± 0.12 g, respectively. A total of 4 different SBC concentrations (0, 0.50, 0.75, and 1 M) were used for injections, and the 0 M group was considered the control group. SBC was obtained from a commercial company that sells food additives [26].

Anesthesia was applied in a 1:9 ratio of clove oil–ethanol solution, which was homogenized. Then, the fish were killed by beating on the head and divided into ‘5 equal regions on both sides of their bodies (both lateral sites of fish).’ Intramuscular SBC injections of 0 M (only pure water), 0.50 M, 0.75 M, and 1 M were applied at 10 different sites (each injection of 2.5 mL; total, 25 mL for 1 fish). A total of 5 fish from each injected group were placed in Styrofoam boxes with 1 row of ice and 1 row of fish to be stored in the refrigerator (4 ± 2 °C) for 0, 24, 48, 72, and 96 h. After the injection procedures, the fish samples were brought to Ordu University, Fatsa Faculty of Marine Sciences, Seafood Processing Technology Laboratories for 0 h analysis, and the other samples were stored in the refrigerator. Proximate composition, physicochemical properties (pH, rigor index, WHC), and physical properties (color and texture profile analyses) were determined at 0, 24, 48, 72, and 96 h of storage.

### 2.2. Proximate Analysis

Protein contents of samples were determined using the Kjeldahl procedure (AOAC, 1998) [27] via a Buchi Digestion System, Model K-424 (BÜCHI Labortechnic, Flawil, Switzerland) and a Kjeltec Distillation Unit B-324 (BÜCHI Labortechnic). A Kjeldahl conversion factor of N × 6.25 was used to calculate the protein proportion. Lipid content was determined using the method described by Bligh and Dyer [28]. Furthermore, ash and moisture analyses were performed according to the methods of AOAC 920.153 [29] and AOAC 950.46 [30], respectively. All details of the analytical methods used have been reported in our previous study [31,32].

### 2.3. Physicochemical Analysis

#### 2.3.1. pH Analysis

We divided both sides of the fish body into 5 equal parts based on visualization from the injection site, and pH analysis from a total of 10 different regions was performed in 3 replications using a digital pH meter (Mettler Toledo pH Meter; Schwerzenbach, Switzerland) according to the report by Santos and others [33].

#### 2.3.2. Determination of Rigor Index

Rigor index of the fish was measured according to the method described by Bito and others [34] and was used as a measure of rigor tension. The fish, immediately after catching and cranial spiking, was placed on a horizontal table, keeping half of the body (tail part) off the table. At selected time intervals, rigor index was calculated using the following formula:IR (&) = [(L_0_ − L_t_)/L_0_] × 100
where L_0_ and L_t_ represent the distances of the base of caudal fin from the horizontal line of the table at the start of the experiment and subsequent storage periods, respectively.

#### 2.3.3. Water Holding Capacity

Water holding capacity was determined according to the method described by Børresen [35]. Water holding capacity of the samples was calculated as percentage remaining water of the initial water content in the sample.
$$WHC=\frac{W_{0}-\Delta W}{W_{0}}\times 100$$
where
$$W_{0}=\frac{V_{0}}{V_{0}+D_{0}}\times 100\;\;\;\;\;\;and\;\;\;\;\;\;\Delta W=\frac{\Delta V_{0}}{V_{0}+D_{0}}\times 100$$

*W*_0_ = initial water content of the sample; Δ*W* = difference in water content of the sample before and after centrifugation; *D*_0_ = initial dry mass of the sample.

### 2.4. Physical Properties

#### 2.4.1. Texture Profile Analysis

The textural parameters of the samples, such as hardness, cohesion, springiness, chewiness, and stickiness, were examined using a Model TA-XT2 tissue analyzer (Stable Micro System, Surrey, UK) fitted with a 2.5 cm diameter roller probe [36].

TPA of samples was conducted using a texture analyzer (TA.XT2i) and the software Texture Expert (v1.20; Stable Micro Systems, Godalming, Surrey, UK) equipped with a load cell of 50 N. Muscle samples with a diameter of 25 mm and a height of 15 ± 3 mm were sampled from fillets. All samples were dried using filter paper after treatment and stored in a refrigerator (4 ± 2 °C) before TPA. The test conditions were 2 successive cycles of compression. A flattened cylindrical plunger (5 mm diameter) was used and pressed into the fillet at a constant speed of 2 mm/s until it reached 30% of the fillet’s depth with 5 s between cycles. The force-time curve was obtained to determine the following parameters: hardness, resilience, springiness, and cohesiveness. Gumminess and chewiness values were calculated by multiplying the hardness and cohesiveness values and the gumminess and springiness values, respectively [37]. The data were processed using Texture Exponent 32 (Stable Micro Systems, Godalming, Surrey, UK). All presented data are means from a triplicate analysis of each sample.

#### 2.4.2. Color Assessment

For color measurements, the CIE *L**, *a**, *b** values of samples were measured using reflectance by a Chroma Meter Konica-Minolta CM-5 (Osaka, Japan) according to Calder [38]. Before starting the analysis, the instrument was calibrated with white and black plates. At least 6 measurements were carried out on each sample. The hue angle and chroma (C*) parameters were calculated as follows:Hue = (*a**^2^ + *b**^2^)^1/2^
Chroma (C*) = Arctan (*b**/*a**)

### 2.5. Statistical Analysis

All the experiments in the study were performed in triplicates. The results are presented as means and standard deviations. Significant differences in the results were determined by applying one-way analysis of variance using SPSS software (version 22; SPSS, Chicago, IL, USA) and the Duncan multiple range test comparisons at a *p* value of <0.05.

## 3. Results and Discussion

### 3.1. Proximate Composition

Protein, lipid, moisture, and ash contents of the rainbow trout were as follows: 17.78 ± 0.33%, 1.39 ± 0.17%, 77.74 ± 0.44%, and 1.14 ± 0.34%. Çelik and Kzak [39] reported no significant difference in the nutritional composition between market-size rainbow trout samples collected from cages and concrete ponds. They observed the highest crude protein ratio (20.63), the highest lipid ratio (2.25), the highest ash ratio (1.39), and the highest dry matter ratio (24.44) in their findings. Similar to our findings, Çelik and others’ [40] moisture, protein, lipid, and ash contents of rainbow trout captured from Atatürk Dam Lake were determined as 71.65, 19.60, 4.43, and 1.36, respectively. Of note, we observed lower lipid contents in our samples compared to those reported in other studies. This distinction has been attributed to the type of food consumed, the presence of high-fat diets, and the restricted activity of farmed fish [41].

### 3.2. Physicochemical Analysis

#### 3.2.1. pH Changes

The changes in the pH of the fish tissue depending on storage duration and the injection concentration are summarized in Table 1. We observed significant associations between the different concentrations of SBC on pH during storage (*p* < 0.05).

The mean pH value of the control, 0.5 M, and 1 M SBC treatment groups did not differ significantly at 0 h (*p* > 0.05). The 0.75 M SBC treatment group had significantly higher pH values at 0 h (*p* < 0.05). We observed a significant increase in the pH values of the treatment groups compared to that of the control group 24 h after the injection. The 0.75 M and 1 M groups had the highest pH values, although insignificant. The lowest and the highest pH were observed in the 0 M and the 0.75 M groups at all storage timepoints, and the difference was significant (*p* < 0.05). The lowest and the highest pH values in the 0 M injection group were 6.549 and 6.789 at 72 h and 0 h, respectively. The lowest pH values for the 0.5 M group were observed at 48 and 72 h of storage. The highest pH for the 0.75 M group was noted at 0 h (*p* < 0.05). The changes in pH values during storage in the 1 M injection group did not differ significantly (*p* > 0.05). Liquid retention is lowest when the muscle pH is close to the isoelectric point (pI) of myofibrillar proteins, whereas moving the pH away from the pI improves liquid retention [42]. NaHCO_3_ is an amphoteric compound with good buffering capacity; aqueous solutions become mildly alkaline, supporting the formation of hydroxide ions (OH^−^) and carbonic acid (H_2_CO_3_), which is predominantly in the form of bicarbonate (HCO_3_^−^) between a pH of 6.4 and 10.3. When used in fish and meat products, NaHCO_3_ is known to raise the muscle pH and hence also alter the protein configuration [43,44]. These results are in line with those reported earlier [17,21]. Other studies with poultry and cattle showed that animals administered SBC showed a significant increase in muscle pH [22,23]. These findings also support those of the current study. González-Rodríguez and others [45] reported increased pH values of trout fillets stored at 3 °C for 10 days from 6.42 to 6.61. Giminez and others [46] also reported similar findings for trout.

The changes in pH by time and injection site are summarized in Table 2. The highest pH value at 0 h was 7.005 at the second region (*p* < 0.05). The variation between the interregional pH values at 24 h was not statistically significant (*p* > 0.05). The highest pH was recorded at 48, 72, and 96 h in the fourth region (*p* < 0.05). The pH did not significantly differ between the first, third, and fourth regions with time (*p* > 0.05).

Table 3 summarizes the pH changes depending on the injection rate applied and the injection site. We did not observe a significant difference in pH between regions in the 0 M and 0.5 M injection groups (*p* > 0.05). In the 0.75 M injection group, the highest pH was detected in the fourth region (*p* < 0.05).

Kang and others [21] reported increased pork pulp pH from 5.68 ± 0.02 to 6.27 ± 0.02 with a 0.42% SBC injection. This difference could be attributed to the better buffer capacity of pork pulp than the pork myofibrillar protein solution. Few studies have reported significant pH increases by adding or increasing the SBC concentration in fish fillets, raw ground beef, raw chicken batter, salted chicken meat, and other similar muscular edibles [47,48].

These results show that SBC increases pH and reduces textural quality losses. The muscle pH increased significantly with increases in the SBC concentration (*p* < 0.05).

#### 3.2.2. Determination of Rigor Index

The onset of rigor mortis is closely associated with the depletion of ATP and glycogen, which is the post-mortem energy state of muscle [49]. Actin and myosin combine to form the actomyosin complex, which promotes irreversible muscle contraction and initiates rigor mortis. Therefore, prolonging the pre-rigor time is an important factor for maximizing fillet yield and shelf life [50]. Evaluation of the rigor index of fish injected with SBC showed that the onset of rigor mortis varied with storage time on ice (Figure 1). Several studies have shown a faster onset of rigor mortis at low pH [50,51]. In the SBC application groups, excluding the control group, the increased muscle pH, depending on the concentration used, caused the entry and exit times to rigor to be prolonged. In particular, the 0.75 and 1 M SBC groups had a significant rigor index at 72 and 96 h. These results and those of previous studies show that these concentrations improve the rigor index by increasing pH and WHC. The rapid introduction of rigor mortis is detrimental to the food processing industry, as filleting fish with full hardness reduces the filet yield and reduces freshness that begins in the post-rigor stage [52].

The results obtained on day 0 are because of the onset of the initial biochemical changes after the death of the fish. Oxygen depletion decreases ATP levels, resulting in sarcomere shortening and subsequent rigor mortis [53]. The hardness and springiness values are the highest and lowest at this stage. Therefore, this study supports our results. The sustained pH drop during rigor mortis promotes the activity of proteolytic enzymes (such as calpain, cathepsins, and proteasome) in fish fillets [54].

#### 3.2.3. Water Holding Capacity

Table 4 summarizes the changes in the WHC in trout muscles. Specifically, we observed an increase in WHC at 0.75% and 1% SBC levels. The WHC differs significantly with storage duration (*p* < 0.05). The 1 M and 0.75 M groups had the highest WHC at 48 h (94.967% and 94.812%, respectively) (*p* < 0.05).

As reported in previous studies supporting our results, SBC use increases the pH, which improves the WHC and textural properties [55]. Studies on the association between the pH and WHC have shown that increasing the pH using phosphates and bicarbonates increases the WHC [56]. SBC is considered a solution to several challenges, such as moisture loss and cooking loss, that occur in products such as ice creams and marinades or with different cooking methods. Fish meat with a higher WHC has better textural properties than that with water loss. Aslı and Morkore [47] found that SBC injection improved tissue hardness in salted cod fillets, as assessed using textural parameters. Alvarado and Sams [57] reported that the functional and flavor properties of pale, tender, and exudative broiler breast meat were improved by adding SBC.

We observed improvements in the textural properties of fish, particularly those injected with 0.75 and 1 M SBC during storage. We noted both the highest pH and WHC values during storage in groups injected with the same SBC concentrations. These results may explain why these concentrations improve textural properties.

In the post-rigor stage, muscle proteins begin to denature, and the WHC of the meat decreases [58]. The characteristics of the post-rigor stage depend on the fish species and temperature. The viscosity and WHC of muscle tissue were significantly lower in sea bass treated with ice-water, and ice-water effectively maintains actin integrity, decreasing muscle damage [59]. In this study, WHC increased significantly with increasing SBC concentrations (*p* < 0.05).

### 3.3. Physical Properties

#### 3.3.1. Texture Profile Analysis

A compression test with a cylindrical probe applied to the seven regions was selected as the most suitable method for textural analysis in rainbow trout (Figure 2). Barroso and others [60] reported differing results obtained from instrumental tissue assessments of salmon muscle, which could be explained by significant differences between individual parts of the fish and sampling point selections. Strength is most commonly used to measure textural quality and to predict the mechanical variation of tissue [61]. However, the results showed that this parameter only distinguishes the tail from the abdomen and back, and the difference between the abdomen and the back was insignificant. The results show that the tail is the hardest region, followed by the abdomen and back.

The effect of the SBC concentration on storage time according to the application areas is summarized in Table 5. In the first zone, the highest hardness value of 12.477 N was noted in the 0.75 M group, and the lowest hardness of 6.224 N was noted in the 0 M group at 96 h. The highest springiness value of 1.800% was noted in the 1 M group on day 0, and the lowest value of 0.964% was noted in the 0.5 M group at 48 h of storage. The highest and the lowest cohesiveness values were 0.819% and 0.678% at day 0 and 96 h, respectively, in the 0 M group. The highest and the lowest gumminess values for the 1 M group were 8141 N at 24 h and 4779 N at 48 h. The highest chewiness value was 19.003 N in the 0.5 M group on day 0, and the lowest was 5.033 N in the 0.75 M group at 48 h. The highest resilience value of 0.621% was noted in the 0 M group at 0 h, and the lowest value of 0.434% was noted for the 0.5 M group at 96 h. Our findings are in line with those reported in the literature. Gao and others [62] reported decreased average hardness, stickiness, chewiness, and springiness values over 15 days of storage at 4 ± 1 °C. Liu and others [63] revealed that the hardness of grass carp fillets stored at −3 °C and 0 °C decreased significantly within the first 3 days. Boughattas and others [8] reported similar values (0.81%–0.90%) from instrumental tissue evaluations of sturgeon fish (*Acipenser gueldenstaedtii*). Zhao and others [64] found that the printability of large yellow croaker fillets under vacuum packaging conditions at 0 °C significantly decreased over a 20-day period.

The highest hardness value in the second region of 17.324 N was observed in the 1 M group at 24 h, and the lowest value of 8.626 N was observed in the 0.5 M group at 96 h. The spring value was 1.483% at the highest 0 M concentration and 0.976% at the lowest 0.75 M concentration after 48 h of storage. The highest cohesiveness value of 0.828% was observed in the 0.5 M group at 0 h, and the lowest value of 0.684% was observed in the 0 M group at 96 h. The highest gumminess value of 13.588 N was observed in the 0 M group at 24 h, and the lowest value of 6.233 N was observed in the 0.5 M group at 96 h. The highest chewiness value of 18.893 N was recorded for the 0 M group at 24 h, and the lowest value of 6.670 N was recorded for the 0.75 M group at 48 h. The highest resilience value of 0.655% was recorded for the 0 M group at day 0, and the lowest value of 0.457% was recorded at 96 h. Lin and others [65] used a compression ratio of 30% to evaluate the instrumental toughness of grass carp fillets (*Ctenopharyngodon idella*) and reported values ranging from 0.39% to 0.49%. In contrast, Iaconisi and others [66] used a compression ratio of 50% to evaluate the textural properties of the black-spotted sea bream (*Pagellus bogaraveo*) and achieved an elasticity value of 0.02%. These findings support the results of our study.

In the third region, the highest hardness value recorded was 21,699 N in the 1 M group at 24 h, and the lowest value recorded was 8032 N in the 0.5 M group at 96 h. The highest springiness value recorded was 1.954% in the 0.75 M group at 0 h, and the lowest value recorded was 0.969% in the 0.5 M group at 48 h. The highest cohesiveness value of 0.870% was recorded in the 0 M group at 0 h, and the lowest value of 0.665% was recorded in the 0 M group at 96 h. The highest gumminess value of 16,532 N was recorded in the 0 M group at 24 h, and the lowest value of 6.332 N was recorded in the 0.5 M group at 96 h. Both the highest (22,028 N) and lowest (7.489 N) chewiness values were recorded in the 0.5 M group, at 0 and at 96 h, respectively. Likewise, the highest (0.693%) and lowest (0.436%) resilience values were recorded in the 0 M group at 0 and 96 h, respectively. Compression decreases cohesiveness, indicating that cohesiveness behaves oppositely to hardness. This finding is in line with that reported in the literature. Wu and others [67] and Tang and others [68] applied 70% and 80% compression, respectively, for the instrumental assessment of fish fillet texture. Wu and others [67] reported results ranging from 0.68% to 0.72%, whereas Tang and others [68] reported results <0.6%.

In the fourth region, the highest hardness value of 28,785 N was recorded in the 0.75 M group at 24 h, and the lowest value of 8115 N was recorded in the 0.5 M group at 96 h. The highest springiness value of 1.864% was recorded in the 0 M group at day 0, and the lowest value of 0.949% was recorded in the 0.5 M group at 48 h. The highest cohesiveness value of 0.841% was recorded in the 0 M group at 0 h, and the lowest value of 0.628% was recorded in the 0.5 M group at 24 h. The highest and lowest gumminess values of 20,779 N and 5706 N were recorded in the 0 M and 0.5 M groups at 24 h and 96 h, respectively. The highest chewiness of 28,044 N was recorded in the 0 M group at day 0, and the lowest value of 5916 N was recorded in the 1 M group at 96 h. The highest and lowest resilience values of 0.671% and 0.412% were recorded in the 0 M group at day 0 and 96 h, respectively. Monteiro and others [6] and Cropotova and others [12] found no significant differences in the instrumental cohesiveness of fish species in the first few days of refrigeration. Hassoun and Karoui [69] reported no significant change in the stickiness of haddock fillets at 4 °C for 15 days.

In the fifth region, the highest hardness value of 29.314 N was recorded in the 1 M group at 24 h, and the lowest value of 8022 N was recorded in the 0.5 M group at 96 h. The highest and lowest springiness values of 1.726% and 0.918% were recorded in the 0.5 M group at 0 h and 0.75 M group at 24 h, respectively. The highest cohesiveness value of 0.830% was observed in the 0.5 M group at 0 h, and the lowest value of 0.627% was observed in the 0 M group at 96 h. The highest gumminess value of 21,046 N was recorded in the 0 M group at 0 h, and the lowest value of 6.038 N was recorded in the 1 M group at 96 h. The highest chewiness value of 30,795 N was observed in the 0 M group at 0 h, and the lowest value of 5670 N was recorded in the 0.5 M group at 96 h. The highest and lowest resilience values of 0.668% and 0.398% were noted in the 0 M group at 0 h and 96 h, respectively. Prolonged storage is associated with advanced protein degradation, primarily affected by cathepsin B and L [54]. These endogenous proteinases degrade the main components of muscle fibers, causing fish fillets to become softer and lose rigidity and elasticity [70,71]. The sensitization process is characterized by a gradual breakdown of the extracellular matrix, whereby the connections between the cytoskeleton and sarcomeres break [72]. Thus, metabolic and structural changes may explain the reduction in hardness values found in the post-mortem period in this study and previous reports [6,8].

In the sixth region, the highest hardness value of 35,652 N was noted in the 1 M group at 48 h, and the lowest value of 10,597 N was observed in the 1 M group at 96 h. The highest springiness value of 1.667% was noted in the 0.5 M group at 0 h, and the lowest value of 0.883% was noted in the 0.75 M group at 48 h. The highest cohesiveness value was 0.811%, as noted in the 0 and 0.5 M group at 0 h, and the lowest value was 0.484%, noted in the 1 M group at 72 h. The highest gumminess value of 24,853 N was found in the 0 M group at 24 h, and the lowest value was 7456 N in the 1 M group at 96 h. The highest chewiness value of 29.100 N was found in the 0.75 M group at 0 h, and the lowest value was 8.079 N, observed in the 0.5 M group at 96 h. Typically, the shelf life of chilled ready fish in the supermarket is approximately 5 days. Li and others [73] reported that the hardness and elasticity of dark and white muscle in common carp (*Cyprinus carpio*) tended to increase and decrease within 72 h of cold storage after slaughter. This finding suggests that the textural quality may deteriorate during refrigerated storage. However, no specific studies have been conducted on the change in textural quality and the mechanism underlying this change during the shelf life of refrigerated fish.

In the seventh region, the highest hardness value of 42.222 N was observed in the 1 M group at 24 h, and the lowest value of 8.097 N was observed in the 0.5 M group at 96 h. The highest springiness value noted was 1.650% in the 0.5 M group at 0 h, and the lowest value was 0.850% in the 0.75 M group at 48 h. The highest cohesiveness value was 0.795%, seen in the 0.5 M group at 0 h, and the lowest was 0.551% in the 0.75 M group at 48 h. The highest gumminess value of 31.715 N was recorded in the 0 M group at 0 h, and the lowest value of 6.322 N was observed in the 0.5 M group at 0 h. The highest chewiness value was 45.698 N, recorded in the 0 M group at 0 h, and the lowest was 9.047 N in the 0.5 M group at 96 h. The highest resilience value was 0.642%, observed at 0 h in the 0.5 M group, and the lowest value was 0.360%, recorded at 48 h in the 0.75 M group. The TPA test showed higher hardness, frangibility, and stickiness values for the tail than for the other regions. However, the stickiness and springiness of the tail sample were lower than those recorded in other regions (*p* < 0.05). Chewiness values were similar across regions, and the tail area was tighter than the rest of the fish’s muscles. Interestingly, several studies have shown that tightness is associated with the density and arrangement of collagen fibers in the connective tissue [74,75]. Dong and others [76] showed that the shear resistance values of trout are higher near the tail because of the higher ratio of insoluble collagen.

#### 3.3.2. Color Assessment

Meat product spoilage is characterized by undesirable changes in color, texture, flavor, and odor. Color is a very important parameter as it is associated with the perception of freshness and quality among consumers. Figure 3 shows the effects of the various concentrations of SBC on color changes (*L**, *a**, *b**) of trout.

*L** value is one of the most important parameters affecting the sensory perception of meat products. Figure 3a shows the changes in *L** values of trout fillets treated with various SBC concentrations during storage. Although the *L** value increased or decreased in all groups during storage, it showed an increasing trend generally. We observed significant differences in *L** at different storage timepoints and between groups. At 0 h of storage, *L** values ranged from 50.00 to 55.00 across groups; we observed the highest *L** values across all groups at 48 to 72 h. Considering the association between the WHC and various concentrations of SBC, the differences in *L** between groups could be explained by the changes in moisture content over the storage duration.

With regard to color parameters, the *a** value is critical to determining the physical quality of fish samples. The ‘+*a**’ value represents redness, and the ‘-*a**’ value represents greenness. Figure 3b shows the changes in *a** values during the storage period. In this study, although the *a** value varied in all application groups during storage, it showed a decreasing trend overall. This decrease in redness value results in an undesirable color [77]. Shabanpour and Etemadian [78] highlighted that adding protein to food products significantly affects the *a** value caused by the heme protein concentration. At 0 h of storage, the control group had the highest *a** value (8.91), and the 1 M group had the lowest *a** value (range, 5.51). The 0.75 M group had the fastest decrease in *a** value.

The *b** value represents yellowness and is crucial to the perceived quality of food products. Figure 3c shows the changes in *b** values of the samples at various SBC concentrations. Variations in *b** values were observed in the control and SBC groups, and these variations continued until the end of storage.

Chroma (C*) is associated with observable differences in the consumption preferences of food products. Figure 3d shows variations and statistical differences in color clarity. At 0 h of storage, the highest and lowest values were observed in the 0 M (21.00) and 0.5 M (16.00) SBC treatment groups, respectively, whereas at the end of the storage, these values were observed in the 1 M (21.00) and 0.75 M (14.00) groups. The chroma value represents color saturation, and a higher chroma value represents color purity [79]. Refrigeration and the sensitivity of fish products to microbial changes may cause a decrease in the chroma value. Cuttle and others [80] proposed that the decrease in chroma value is associated with bacterial activity.

The hue value (H*), another color parameter, represents the color tones of the products and indicates quality losses when comparing the initial and final values. Figure 3e shows the changes in the color tone of samples stored for 96 h. The initial color tone values ranged from 65 to 72 at 0 h and decreased to 68–80 at 96 h.

## 4. Conclusions

The results of our study showed improved textural properties of fish treated with SBC, particularly with 0.75 and 1 M SBC. The same SBC groups had the highest pH and WHC values during storage. These findings may explain why these SBC concentrations improve textural properties. In addition, the muscle’s chemical composition and physical structure change along the fish’s body line and affect its textural properties. The TPA test showed higher hardness, frangibility, and stickiness values for the tail than for the other regions. However, the stickiness and springiness of the tail samples were lower than those recorded in other regions. Increased muscle pH at 0.75 and 1 M SBC concentrations prolonged the entry and exit times in rigor.

## Figures and Tables

**Figure 1 foods-12-02437-f001:**
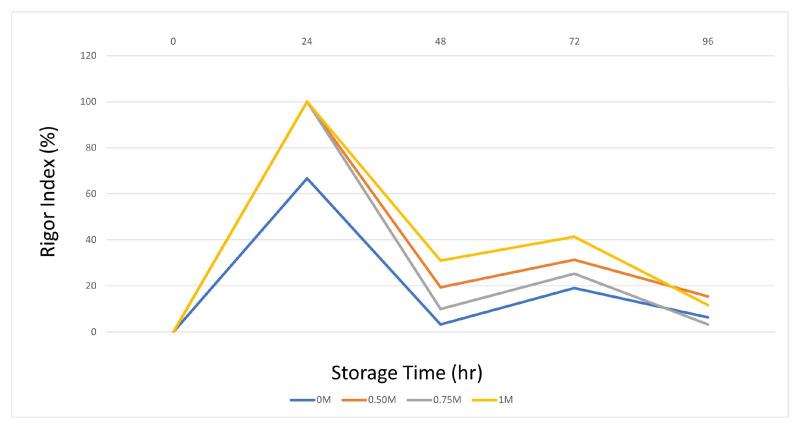
Rigor index changes.

**Figure 2 foods-12-02437-f002:**
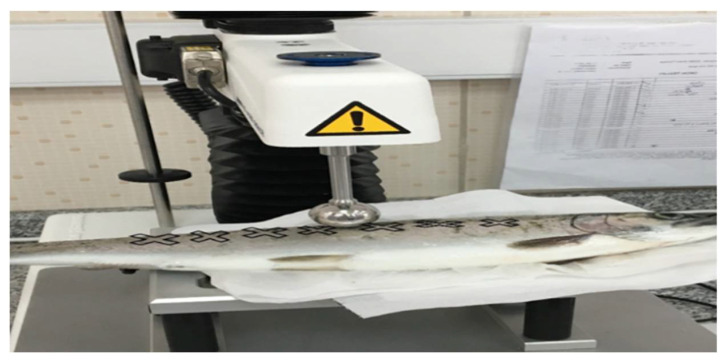
Textural measurements regions of SBC-injected samples.

**Figure 3 foods-12-02437-f003:**
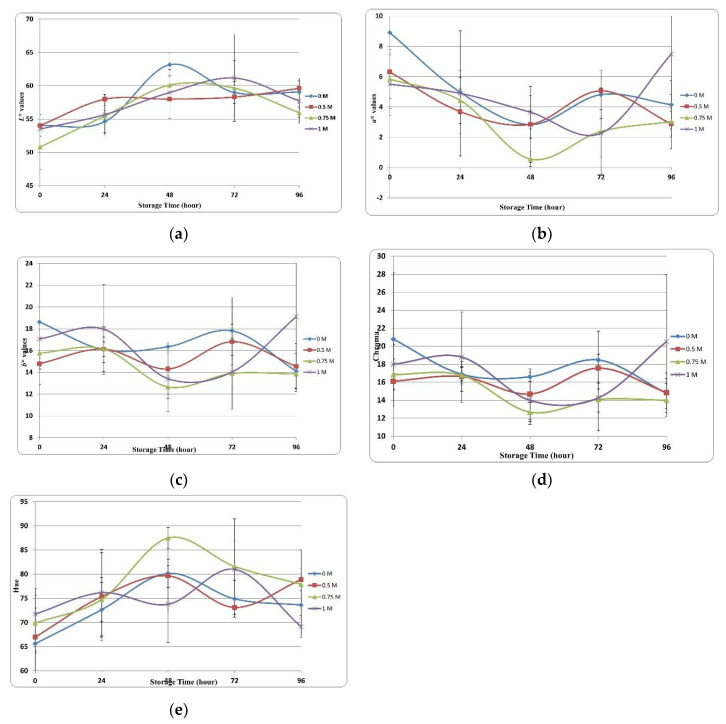
Color changes of SBC injected samples. (**a**) *L** values. (**b**) *a** values. (**c**) *b** values. (**d**) *Chroma* values. (**e**) Hue values.

**Table 1 foods-12-02437-t001:** pH changes in SBC-injected samples (hour/molarite).

Hour	Molarite (M)			
	0 M	0.50 M	0.75 M	1 M
**0**	6.789 ± 0.273 ^az^	6.817 ± 0.440 ^ay^	7.051 ± 0.500 ^by^	6.869 ± 0.195 ^ax^
**24**	6.590 ± 0.700 ^axy^	6.754 ± 0.196 ^bxy^	6.818 ± 0.295 ^bx^	6.757 ± 0.228 ^bx^
**48**	6.624 ± 0.848 ^axy^	6.675 ± 0.211 ^ax^	6.949 ± 0.265 ^cxy^	6.775 ± 0.154 ^bx^
**72**	6.549 ± 0.048 ^ax^	6.673 ± 0.179 ^bx^	6.856 ± 0.211 ^cx^	6.796 ± 0.143 ^cx^
**96**	6.646 ± 0.101 ^ay^	6.681 ± 0.114 ^abxy^	6.856 ± 0.207 ^cx^	6.747 ± 0.219 ^bx^

a,b,c: The difference between averages with different letters on the same line is significant (*p* < 0.05). x,y,z: The difference between means with different numbers in the same column is significant (*p* < 0.05).

**Table 2 foods-12-02437-t002:** pH changes of SBC-injected samples (hour/region).

Hour	Region				
	1	2	3	4	5
**0**	6.738 ± 0.266 ^ax^	7.005 ± 0.488 ^by^	6.865 ± 0.387 ^abx^	6.870 ± 0.354 ^abx^	6.868 ± 0.374 ^aby^
**24**	6.665 ± 0.230 ^ax^	6.709 ± 0.256 ^ax^	6.712 ± 0.201 ^ax^	6.795 ± 0.243 ^ax^	6.768 ± 0.196 ^axy^
**48**	6.674 ± 0.149 ^ax^	6.703 ± 0.171 ^ax^	6.736 ± 0.185 ^abx^	6.857 ± 0.291 ^bx^	6.810 ± 0.264 ^abxy^
**72**	6.649 ± 0.142 ^ax^	6.653 ± 0.152 ^ax^	6.760 ± 0.210 ^abx^	6.788 ± 0.245 ^bx^	6.744 ± 0.184 ^abxy^
**96**	6.711 ± 0.148 ^abx^	6.678 ± 0.124 ^ax^	6.778 ± 0.196 ^abx^	6.812 ± 0.245 ^bx^	6.683 ± 0.162 ^ax^

a,b: The difference between averages with different letters on the same line is significant (*p* < 0.05). x,y: The difference between means with different numbers in the same column is significant (*p* < 0.05).

**Table 3 foods-12-02437-t003:** pH changes of SBC-injected samples (molarite/region).

	Region				
Molarite (M)	1	2	3	4	5
**0**	6.643 ± 0.229 ^ax^	6.650 ± 0.196 ^ax^	6.619 ± 0.104 ^ax^	6.635 ± 0.145 ^ax^	6.651 ± 0.100 ^ax^
**0.5**	6.643 ± 0.137 ^ax^	6.756 ± 0.429 ^axy^	6.745 ± 0.215 ^ay^	6.739 ± 0.196 ^ax^	6.719 ± 0.207 ^axy^
**0.75**	6.741 ± 0.195 ^ax^	6.686 ± 0.271 ^abxy^	6.957 ± 0.340 ^bz^	7.029 ± 0.344 ^bz^	6.940 ± 0.370 ^bz^
**1**	6.722 ± 0.189 ^ax^	6.729 ± 0.208 ^axy^	6.759 ± 0.148 ^ay^	6.895 ± 0.204 ^by^	6.789 ± 0.151 ^ay^

a,b: The difference between averages with different letters on the same line is significant (*p* < 0.05). x,y,z: The difference between means with different numbers in the same column is significant (*p* < 0.05).

**Table 4 foods-12-02437-t004:** WHC changes in SBC-injected samples.

Hour	Molarite (M)			
	0	0.50	0.75	1
**0**	89.837 ± 2.164 ^ax^	89.766 ± 1.318 ^ax^	90.476 ± 1.2446 ^bx^	90.078 ± 0.748 ^abx^
**24**	92.990 ± 3.781 ^axy^	93.581 ± 1.639 ^abxy^	93.901 ± 1.176 ^abxy^	94.360 ± 1.970 ^by^
**48**	91.443 ± 3.482 ^ax^	94.706 ± 1.985 ^aby^	94.812 ± 3.485 ^aby^	94.967 ± 5.582 ^by^
**72**	90.754 ± 0.324 ^ax^	91.855 ± 4.914 ^abx^	92.690 ± 2.949 ^bxy^	91.209 ± 5.175 ^abx^
**96**	90.652 ± 2.323 ^ax^	93.268 ± 6.628 ^bxy^	93.790 ± 4.379 ^bxy^	92.453 ± 4.325 ^abxy^

a,b: The difference between averages with different letters on the same line is significant (*p* < 0.05). x,y: The difference between means with different numbers in the same column is significant (*p* < 0.05).

**Table 5 foods-12-02437-t005:** Texture changes of SBC-injected samples.

Concentration	Hardness (Newton) (N)	Springiness (Ratio) (%)	Cohesiveness (Ratio) (%)	Gumminess (Newton) (N)	Chewiness (Newton) (N)	Resilience (Ratio) (%)	Storage Hours
	**1. Region**						
**0**	8.665 ± 2.121 ^a^	1.577 ± 0.518 ^a^	0.819 ± 0.043 ^b^	7.144 ± 1.553 ^a^	12.527 ± 1.673 ^b^	0.621 ± 0.019 ^c^	**0**
	11.978 ± 0.926 ^b^	1.231 ± 0.419 ^a^	0.710 ± 0.037 ^a^	12.871 ± 0.798 ^b^	14.808 ± 1.737 ^b^	0.552 ± 0.051 ^b^	**24**
	9.843 ± 1.052 ^a^	1.279 ± 0.256 ^a^	0.731 ± 0.019 ^a^	7.205 ± 0.881 ^a^	9.068 ± 0.821 ^a^	0.497 ± 0.003 ^ab^	**48**
	9.780 ± 0.784 ^a^	0.993 ± 0.008 ^a^	0.721 ± 0.024 ^a^	7.587 ± 1.235 ^a^	7.530 ± 1.164 ^a^	0.474 ± 0.006 ^a^	**72**
	6.224 ± 1.057 ^a^	1.289 ± 0.519 ^a^	0.678 ± 0.061 ^a^	7.095 ± 0.151 ^a^	7.705 ± 1.289 ^a^	0.443 ± 0.041 ^a^	**96**
**0.5**	12.221 ± 1.220 ^c^	1.780 ± 0.323 ^b^	0.807 ± 0.019 ^b^	9.831 ± 1.007 ^c^	19.003 ± 2.569 ^b^	0.615 ± 0.031 ^c^	**0**
	11.044 ± 1.531 ^bc^	1.021 ± 0.046 ^a^	0.746 ± 0.037 ^a^	8.364 ± 1.066 ^bc^	8.599 ± 0.818 ^a^	0.525 ± 0.026 ^b^	**24**
	10.733 ± 0.966 ^bc^	0.964 ± 0.043 ^a^	0.705 ± 0.043 ^a^	8.120 ± 1.075 ^bc^	7.798 ± 0.678 ^a^	0.470 ± 0.023 ^a^	**48**
	9.152 ± 1.195 ^ab^	1.117 ± 0.306 ^a^	0.733 ± 0.024 ^a^	6.690 ± 0.702 ^ab^	6.687 ± 0.965 ^a^	0.471 ± 0.018 ^a^	**72**
	6.294 ± 1.478 ^a^	1.150 ± 0.290 ^a^	0.696 ± 0.031 ^a^	5.396 ± 0.829 ^a^	6.043 ± 0.488 ^a^	0.434 ± 0.021 ^a^	**96**
**0.75**	7.769 ± 0.959 ^a^	1.766 ± 0.343 ^b^	0.793 ± 0.020 ^c^	6.149 ± 0.607 ^a^	11.816 ± 1.302 ^c^	0.582 ± 0.036 ^c^	**0**
	12.477 ± 1.084 ^b^	1.059 ± 0.139 ^a^	0.743 ± 0.034 ^b^	8.903 ± 0.487 ^b^	9.384 ± 0.752 ^b^	0.524 ± 0.036 ^bc^	**24**
	7.272 ± 1.831 ^a^	0.982 ± 0.017 ^a^	0.710 ± 0.017 ^ab^	5.128 ± 1.270 ^a^	5.033 ± 1.305 ^a^	0.448 ± 0.036 ^a^	**48**
	9.194 ± 1.285 ^a^	1.001 ± 0.013 ^a^	0.741 ± 0.018 ^b^	6.830 ± 1.096 ^a^	6.846 ± 1.172 ^a^	0.478 ± 0.028 ^ab^	**72**
	9.744 ± 1.124 ^ab^	0.988 ± 0.007 ^a^	0.699 ± 0.015 ^a^	6.338 ± 1.537 ^a^	6.269 ± 1.557 ^a^	0.439 ± 0.024 ^a^	**96**
**1**	9.900 ± 0.670 ^b^	1.800 ± 0.184 ^b^	0.754 ± 0.023 ^ab^	7.494 ± 0.474 ^b^	14.297 ± 1.261 ^b^	0.572 ± 0.025 ^b^	**0**
	11.693 ± 2.121 ^b^	1.069 ± 0.132 ^a^	0.749 ± 0.024 ^ab^	8.141 ± 0.445 ^b^	7.694 ± 0.659 ^a^	0.539 ± 0.012 ^b^	**24**
	6.607 ± 1.804 ^a^	1.210 ± 0.285 ^ab^	0.715 ± 0.055 ^ab^	4.779 ± 1.126 ^a^	7.580 ± 0.803 ^a^	0.457 ± 0.009 ^a^	**48**
	10.564 ± 1.201 ^b^	1.197 ± 0.384 ^ab^	0.691 ± 0.029 ^a^	6.713 ± 1.474 ^b^	7.671 ± 0.610 ^a^	0.450 ± 0.020 ^a^	**72**
	6.628 ± 0.267 ^a^	1.292 ± 0.517 ^ab^	0.759 ± 0.025 ^b^	5.029 ± 1.474 ^a^	6.530 ± 2.765 ^a^	0.477 ± 0.049 ^a^	**96**
	**2. Region**						
**0**	10.893 ± 1.382 ^a^	1.650 ± 0.569 ^a^	0.827 ± 0.048 ^c^	8.428 ± 1.612 ^a^	15.064 ± 0.90 ^b^	0.655 ± 0.057 ^b^	**0**
	16.522 ± 0.624 ^b^	1.254 ± 0.456 ^a^	0.776 ± 0.042 ^bc^	13.588 ± 1.139 ^b^	18.893 ± 0.906 ^c^	0.606 ± 0.043 ^b^	**24**
	11.637 ± 0.796 ^a^	1.483 ± 0.444 ^a^	0.734 ± 0.010 ^ab^	8.538 ± 0.480 ^a^	13.904 ± 1.154 ^b^	0.518 ± 0.024 ^a^	**48**
	10.788 ± 0.696 ^a^	1.268 ± 0.480 ^a^	0.750 ± 0.032 ^ab^	8.198 ± 0.390 ^a^	10.060 ± 0.657 ^a^	0.508 ± 0.021 ^a^	**72**
	11.737 ± 0.814 ^a^	1.308 ± 0.518 ^a^	0.684 ± 0.046 ^a^	8.021 ± 0.620 ^a^	9.312 ± 1.003 ^a^	0.457 ± 0.041 ^a^	**96**
**0.5**	10.559 ± 0.782 ^b^	1.976 ± 0.025 ^b^	0.828 ± 0.030 ^b^	9.437 ± 1.574 ^b^	17.422 ± 1.070 ^d^	0.644 ± 0.031 ^c^	**0**
	15.277 ± 1.029 ^d^	1.265 ± 0.492 ^a^	0.740 ± 0.020 ^a^	10.182 ± 0.962 ^b^	11.850 ± 1.329 ^c^	0.535 ± 0.012 ^b^	**24**
	12.587 ± 1.061 ^c^	0.984 ± 0.021 ^a^	0.756 ± 0.031 ^a^	9.159 ± 0.755 ^b^	9.003 ± 0.578 ^b^	0.529 ± 0.022 ^b^	**48**
	9.253 ± 0.678 ^ab^	1.009 ± 0.050 ^a^	0.756 ± 0.018 ^a^	7.040 ± 0.500 ^a^	7.917 ± 0.694 ^ab^	0.500 ± 0.021 ^ab^	**72**
	8.626 ± 0.748 ^a^	1.259 ± 0.465 ^a^	0.723 ± 0.009 ^a^	6.233 ± 0.472 ^a^	6.832 ± 0.842 ^a^	0.463 ± 0.009 ^a^	**96**
**0.75**	9.797 ± 1.465 ^a^	1.612 ± 0.361 ^b^	0.808 ± 0.010 ^b^	7.916 ± 1.188 ^ab^	13.584 ± 0.906 ^c^	0.613 ± 0.031 ^b^	**0**
	15.298 ± 1.006 ^b^	1.267 ± 0.455 ^ab^	0.768 ± 0.030 ^ab^	12.843 ± 0.710 ^b^	14.170 ± 1.289 ^c^	0.570 ± 0.037 ^b^	**24**
	10.298 ± 0.777 ^a^	0.976 ± 0.051 ^a^	0.719 ± 0.051 ^a^	6.294 ± 1.066 ^a^	6.670 ± 0.428 ^a^	0.471 ± 0.024 ^a^	**48**
	10.724 ± 0.496 ^a^	1.069 ± 0.129 ^a^	0.757 ± 0.008 ^ab^	8.123 ± 0.380 ^b^	8.458 ± 0.379 ^b^	0.505 ± 0.024 ^a^	**72**
	10.710 ± 1.076 ^a^	1.071 ± 0.125 ^a^	0.723 ± 0.013 ^a^	7.738 ± 0.782 ^ab^	8.224 ± 0.095 ^b^	0.473 ± 0.007 ^a^	**96**
**1**	10.864 ± 0.854 ^bc^	1.910 ± 0.107 ^b^	0.784 ± 0.002 ^a^	8.513 ± 0.654 ^b^	16.556 ± 0.936 ^c^	0.610 ± 0.008 ^c^	**0**
	17.324 ± 1.087 ^d^	1.059 ± 0.119 ^a^	0.744 ± 0.043 ^a^	12.643 ± 0.825 ^c^	12.429 ± 0.973 ^b^	0.555 ± 0.034 ^b^	**24**
	9.852 ± 0.737 ^ab^	1.251 ± 0.335 ^ab^	0.728 ± 0.070 ^a^	7.267 ± 0.402 ^a^	10.096 ± 0.688 ^ab^	0.490 ± 0.034 ^a^	**48**
	12.246 ± 1.159 ^c^	1.279 ± 0.539 ^ab^	0.719 ± 0.022 ^a^	8.790 ± 0.702 ^b^	9.528 ± 1.154 ^ab^	0.488 ± 0.016 ^a^	**72**
	8.353 ± 0.930 ^a^	1.319 ± 0.531 ^ab^	0.753 ± 0.007 ^a^	6.286 ± 0.652 ^a^	8.305 ± 3.482 ^a^	0.494 ± 0.022 ^a^	**96**
	**3. Region**						
**0**	9.978 ± 0.702 ^a^	1.659 ± 0.299 ^a^	0.870 ± 0.062 ^c^	8.644 ± 0.536 ^ab^	11.386 ± 0.826 ^b^	0.693 ± 0.071 ^c^	**0**
	21.530 ± 1.488 ^c^	1.101 ± 0.194 ^a^	0.760 ± 0.016 ^b^	16.532 ± 1.222 ^c^	18.103 ± 1.790 ^d^	0.581 ± 0.050 ^b^	**24**
	12.432 ± 0.559 ^b^	1.482 ± 0.476 ^a^	0.732 ± 0.028 ^ab^	9.543 ± 0.864 ^b^	14.648 ± 0.687 ^c^	0.520 ± 0.048 ^ab^	**48**
	12.402 ± 0.679 ^b^	1.151 ± 0.262 ^a^	0.726 ± 0.009 ^ab^	8.662 ± 0.855 ^ab^	9.349 ± 0.595 ^a^	0.481 ± 0.007 ^b^	**72**
	11.161 ± 0.345 ^ab^	1.294 ± 0.525 ^a^	0.665 ± 0.065 ^a^	7.420 ± 0.714 ^a^	8.416 ± 1.152 ^a^	0.436 ± 0.053 ^b^	**96**
**0.5**	12.636 ± 0.427 ^c^	1.946 ± 0.030 ^b^	0.857 ± 0.055 ^b^	11.505 ± 0.569 ^c^	22.028 ± 0.544 ^d^	0.686 ± 0.064 ^b^	**0**
	21.201 ± 1.064 ^e^	1.264 ± 0.328 ^a^	0.720 ± 0.010 ^a^	15.227 ± 0.714 ^d^	18.392 ± 0.960 ^c^	0.529 ± 0.008 ^a^	**24**
	15.189 ± 0.721 ^d^	0.969 ± 0.019 ^a^	0.730 ± 0.060 ^a^	10.408 ± 1.000 ^c^	10.078 ± 0.859 ^b^	0.508 ± 0.056 ^a^	**48**
	10.016 ± 0.747 ^b^	1.135 ± 0.307 ^a^	0.753 ± 0.026 ^a^	7.929 ± 0.906 ^b^	8.241 ± 0.610 ^a^	0.493 ± 0.027 ^a^	**72**
	8.032 ± 0.629 ^a^	1.291 ± 0.536 ^a^	0.727 ± 0.033 ^a^	6.332 ± 1.076 ^a^	7.489 ± 0.528 ^a^	0.464 ± 0.030 ^a^	**96**
**0.75**	10.772 ± 0.743 ^a^	1.954 ± 0.036 ^b^	0.836 ± 0.044 ^b^	8.527 ± 0.816 ^ab^	17.158 ± 0.648 ^d^	0.645 ± 0.057 ^c^	**0**
	18.841 ± 0.239 ^c^	1.276 ± 0.496 ^a^	0.748 ± 0.046 ^a^	14.100 ± 0.886 ^c^	14.016 ± 0.955 ^c^	0.558 ± 0.060 ^b^	**24**
	13.301 ± 0.670 ^b^	1.034 ± 0.054 ^a^	0.697 ± 0.016 ^a^	9.319 ± 0.418 ^b^	9.110 ± 0.883 ^b^	0.472 ± 0.025 ^a^	**48**
	11.524 ± 0.749 ^a^	1.072 ± 0.148 ^a^	0.725 ± 0.005 ^a^	8.350 ± 0.483 ^ab^	8.903 ± 0.686 ^ab^	0.480 ± 0.017 ^a^	**72**
	10.292 ± 0.879 ^a^	1.059 ± 0.109 ^a^	0.703 ± 0.010 ^a^	7.235 ± 0.693 ^a^	7.626 ± 0.560 ^a^	0.457 ± 0.011 ^a^	**96**
**1**	11.500 ± 0.817 ^bc^	1.507 ± 0.402 ^a^	0.779 ± 0.011 ^b^	8.919 ± 0.683 ^b^	15.680 ± 1.019 ^c^	0.608 ± 0.015 ^c^	**0**
	21.399 ± 1.423 ^d^	0.994 ± 0.014 ^a^	0.710 ± 0.031 ^a^	15.017 ± 0.769 ^c^	14.849 ± 0.730 ^c^	0.525 ± 0.015 ^b^	**24**
	12.383 ± 0.494 ^c^	1.421 ± 0.392 ^a^	0.708 ± 0.043 ^a^	8.960 ± 0.622 ^b^	12.401 ± 0.986 ^ab^	0.485 ± 0.026 ^ab^	**48**
	10.247 ± 0.739 ^ab^	1.268 ± 0.495 ^a^	0.715 ± 0.025 ^a^	7.353 ± 0.574 ^a^	9.333 ± 0.474 ^ab^	0.478 ± 0.007 ^a^	**72**
	8.857 ± 0.965 ^a^	1.306 ± 0.541 ^a^	0.732 ± 0.036 ^ab^	6.471 ± 0.536 ^a^	8.589 ± 4.147 ^a^	0.472 ± 0.038 ^a^	**96**
	**4. Region**						
**0**	18.106 ± 1.862 ^c^	1.864 ± 0.104 ^b^	0.841 ± 0.073 ^c^	16.188 ± 1.601 ^d^	28.044 ± 1.231 ^e^	0.671 ± 0.077 ^c^	**0**
	25.006 ± 1.528 ^d^	0.990 ± 0.002 ^a^	0.761 ± 0.040 ^bc^	20.779 ± 1.309 ^e^	19.088 ± 1.306 ^d^	0.574 ± 0.051 ^b^	**24**
	15.446 ± 0.630 ^b^	1.212 ± 0.417 ^a^	0.723 ± 0.021 ^ab^	11.567 ± 1.119 ^c^	13.332 ± 1.241 ^c^	0.507 ± 0.040 ^ab^	**48**
	13.986 ± 0.592 ^b^	0.999 ± 0.018 ^a^	0.678 ± 0.007 ^ab^	9.450 ± 0.444 ^b^	9.384 ± 0.516 ^b^	0.440 ± 0.004 ^a^	**72**
	10.977 ± 0.797 ^a^	1.033 ± 0.092 ^a^	0.647 ± 0.071 ^a^	7.091 ± 0.778 ^a^	7.324 ± 0.978 ^a^	0.412 ± 0.056 ^a^	**96**
**0.5**	18.421 ± 0.939 ^c^	1.646 ± 0.587 ^a^	0.805 ± 0.052 ^c^	14.453 ± 0.699 ^c^	18.906 ± 1.656 ^c^	0.629 ± 0.067 ^b^	**0**
	28.637 ± 0.892 ^e^	0.960 ± 0.021 ^a^	0.628 ± 0.050 ^a^	18.239 ± 0.729 ^d^	17.537 ± 0.744 ^c^	0.443 ± 0.050 ^a^	**24**
	22.168 ± 0.727 ^d^	0.949 ± 0.034 ^a^	0.665 ± 0.050 ^ab^	14.647 ± 0.590 ^c^	13.896 ± 0.372 ^b^	0.464 ± 0.055 ^a^	**48**
	12.111 ± 0.728 ^b^	1.167 ± 0.386 ^a^	0.720 ± 0.032 ^b^	8.353 ± 0.642 ^a^	8.651 ± 0.823 ^a^	0.462 ± 0.034 ^a^	**72**
	8.115 ± 0.674 ^a^	1.164 ± 0.365 ^a^	0.702 ± 0.019 ^ab^	5.706 ± 0.447 ^a^	7.776 ± 0.821 ^a^	0.435 ± 0.024 ^a^	**96**
**0.75**	11.854 ± 0.805 ^b^	1.555 ± 0.490 ^b^	0.814 ± 0.040 ^c^	9.471 ± 0.745 ^b^	14.630 ± 1.171 ^c^	0.622 ± 0.052 ^c^	**0**
	28.785 ± 1.917 ^d^	1.118 ± 0.260 ^ab^	0.717 ± 0.045 ^b^	20.101 ± 1.537 ^d^	22.866 ± 0.751 ^d^	0.530 ± 0.056 ^b^	**24**
	17.217 ± 0.761 ^c^	0.976 ± 0.014 ^a^	0.649 ± 0.029 ^a^	11.210 ± 0.349 ^c^	10.895 ± 0.315 ^b^	0.430 ± 0.010 ^a^	**48**
	10.442 ± 0.921 ^ab^	0.988 ± 0.006 ^a^	0.726 ± 0.020 ^b^	7.578 ± 0.725 ^a^	7.483 ± 0.689 ^a^	0.466 ± 0.011 ^a^	**72**
	10.175 ± 0.431 ^a^	0.963 ± 0.047 ^a^	0.690 ± 0.006 ^ab^	6.503 ± 0.939 ^a^	6.243 ± 0.279 ^a^	0.436 ± 0.009 ^a^	**96**
**1**	16.019 ± 1.588 ^c^	1.589 ± 0.246 ^a^	0.761 ± 0.008 ^b^	12.209 ± 1.276 ^d^	19.210 ± 1.387 ^d^	0.594 ± 0.019 ^b^	**0**
	25.632 ± 1.048 ^d^	0.973 ± 0.008 ^a^	0.663 ± 0.034 ^a^	16.984 ± 0.404 ^e^	16.524 ± 0.270 ^c^	0.478 ± 0.037 ^a^	**24**
	17.149 ± 0.815 ^c^	1.109 ± 0.211 ^a^	0.652 ± 0.050 ^a^	10.740 ± 0.239 ^c^	11.498 ± 0.930 ^b^	0.440 ± 0.039 ^a^	**48**
	11.499 ± 0.951 ^b^	1.245 ± 0.549 ^a^	0.676 ± 0.017 ^a^	7.772 ± 0.539 ^b^	10.452 ± 0.913 ^b^	0.445 ± 0.017 ^a^	**72**
	8.486 ± 0.667 ^a^	1.322 ± 0.559 ^a^	0.736 ± 0.031 ^b^	6.024 ± 0.748 ^a^	5.916 ± 0.604 ^a^	0.460 ± 0.024 ^a^	**96**
	**5. Region**						
**0**	26.786 ± 1.071 ^c^	1.510 ± 0.494 ^a^	0.817 ± 0.081 ^c^	21.046 ± 0.752 ^d^	30.795 ± 1.742 ^d^	0.668 ± 0.087 ^c^	**0**
	19.460 ± 0.853 ^b^	1.553 ± 0.503 ^a^	0.775 ± 0.076 ^bc^	15.117 ± 0.848 ^c^	26.950 ± 0.730 ^c^	0.593 ± 0.079 ^bc^	**24**
	20.890 ± 1.124 ^b^	0.978 ± 0.049 ^a^	0.690 ± 0.041 ^abc^	14.381 ± 0.241 ^c^	14.053 ± 0.523 ^b^	0.487 ± 0.053 ^ab^	**48**
	18.922 ± 1.480 ^b^	1.170 ± 0.349 ^a^	0.660 ± 0.036 ^ab^	12.185 ± 0.812 ^b^	12.764 ± 1.002 ^b^	0.431 ± 0.029 ^a^	**72**
	11.916 ± 0.683 ^a^	1.255 ± 0.452 ^a^	0.627 ± 0.098 ^a^	7.092 ± 1.004 ^a^	8.925 ± 0.893 ^a^	0.398 ± 0.084 ^a^	**96**
**0.5**	17.025 ± 1.995 ^c^	1.726 ± 0.276 ^b^	0.830 ± 0.050 ^b^	13.628 ± 1.487 ^c^	20.440 ± 0.885 ^d^	0.662 ± 0.067 ^b^	**0**
	28.628 ± 0.983 ^e^	0.950 ± 0.069 ^a^	0.655 ± 0.055 ^a^	16.832 ± 0.910 ^d^	14.474 ± 0.832 ^c^	0.455 ± 0.046 ^a^	**24**
	26.192 ± 0.614 ^d^	0.920 ± 0.061 ^a^	0.650 ± 0.063 ^a^	16.580 ± 0.974 ^d^	15.226 ± 0.868 ^c^	0.447 ± 0.069 ^a^	**48**
	13.813 ± 0.991 ^b^	1.056 ± 0.158 ^a^	0.705 ± 0.021 ^a^	9.331 ± 0.905 ^b^	9.758 ± 0.519 ^b^	0.441 ± 0.030 ^a^	**72**
	8.022 ± 0.969 ^a^	1.009 ± 0.027 ^a^	0.705 ± 0.006 ^a^	6.380 ± 0.719 ^a^	5.670 ± 0.519 ^a^	0.430 ± 0.014 ^a^	**96**
**0.75**	17.347 ± 1.272 ^b^	1.554 ± 0.166 ^b^	0.787 ± 0.014 ^b^	13.734 ± 1.033 ^b^	21.254 ± 1.486 ^c^	0.603 ± 0.038 ^b^	**0**
	22.898 ± 2.470 ^c^	0.918 ± 0.026 ^a^	0.678 ± 0.087 ^a^	14.423 ± 1.353 ^b^	13.049 ± 1.202 ^b^	0.465 ± 0.059 ^a^	**24**
	20.825 ± 2.106 ^c^	0.943 ± 0.067 ^a^	0.630 ± 0.011 ^a^	13.010 ± 1.277 ^b^	12.112 ± 1.776 ^b^	0.419 ± 0.009 ^a^	**48**
	11.815 ± 0.897 ^a^	0.986 ± 0.022 ^a^	0.705 ± 0.011 ^a^	8.277 ± 0.587 ^a^	8.101 ± 0.527 ^a^	0.454 ± 0.013 ^a^	**72**
	8.975 ± 0.501 ^a^	0.993 ± 0.002 ^a^	0.692 ± 0.005 ^a^	6.614 ± 1.079 ^a^	6.564 ± 1.055 ^a^	0.427 ± 0.007 ^a^	**96**
**1**	21.452 ± 2.089 ^c^	1.105 ± 0.209 ^a^	0.740 ± 0.011 ^b^	14.103 ± 1.284 ^cd^	16.903 ± 0.598 ^e^	0.560 ± 0.041 ^c^	**0**
	29.314 ± 0.857 ^d^	0.975 ± 0.022 ^a^	0.685 ± 0.026 ^ab^	15.242 ± 0.569 ^d^	14.884 ± 0.584 ^d^	0.498 ± 0.020 ^bc^	**24**
	20.562 ± 1.834 ^c^	0.973 ± 0.043 ^a^	0.630 ± 0.056 ^a^	12.712 ± 0.448 ^c^	11.860 ± 1.126 ^c^	0.425 ± 0.045 ^a^	**48**
	13.187 ± 0.684 ^b^	1.145 ± 0.408 ^a^	0.670 ± 0.022 ^a^	9.189 ± 1.015 ^b^	9.105 ± 0.853 ^b^	0.431 ± 0.041 ^ab^	**72**
	8.237 ± 0.875 ^a^	1.318 ± 0.557 ^a^	0.692 ± 0.033 ^ab^	6.038 ± 1.038 ^a^	6.842 ± 1.321 ^a^	0.436 ± 0.032 ^ab^	**96**
	**6. Region**						
**0**	31.552 ± 1.256 ^d^	1.394 ± 0.297 ^b^	0.811 ± 0.086 ^c^	24.102 ± 0.712 ^d^	28.529 ± 2.722 ^d^	0.667 ± 0.093 ^c^	**0**
	32.398 ± 1.995 ^d^	1.008 ± 0.033 ^a^	0.756 ± 0.049 ^bc^	24.853 ± 1.309 ^d^	25.290 ± 1.504 ^c^	0.580 ± 0.066 ^bc^	**24**
	20.890 ± 1.124 ^c^	0.962 ± 0.055 ^a^	0.675 ± 0.029 ^ab^	17.652 ± 0.595 ^c^	17.236 ± 0.472 ^b^	0.488 ± 0.045 ^ab^	**48**
	18.922 ± 1.480 ^b^	0.941 ± 0.076 ^a^	0.609 ± 0.053 ^a^	13.836 ± 0.951 ^b^	11.727 ± 0.588 ^a^	0.398 ± 0.047 ^a^	**72**
	11.916 ± 0.683 ^a^	1.062 ± 0.134 ^a^	0.600 ± 0.100 ^a^	8.991 ± 0.628 ^a^	9.823 ± 1.269 ^a^	0.384 ± 0.096 ^a^	**96**
**0.5**	17.897 ± 0.906 ^b^	1.667 ± 0.498 ^b^	0.811 ± 0.019 ^b^	14.350 ± 0.708 ^c^	23.073 ± 1.188 ^e^	0.645 ± 0.044 ^b^	**0**
	25.016 ± 1.900 ^c^	1.130 ± 0.313 ^a^	0.699 ± 0.024 ^a^	17.245 ± 1.247 ^d^	16.932 ± 0.764 ^d^	0.495 ± 0.000 ^a^	**24**
	23.552 ± 1.623 ^c^	0.931 ± 0.117 ^a^	0.663 ± 0.117 ^a^	16.640 ± 0.477 ^d^	14.300 ± 0.823 ^c^	0.455 ± 0.099 ^a^	**48**
	16.266 ± 0.854 ^b^	0.979 ± 0.040 ^a^	0.705 ± 0.027 ^a^	11.321 ± 0.770 ^b^	11.230 ± 0.573 ^b^	0.445 ± 0.007 ^a^	**72**
	12.122 ± 0.686 ^a^	1.073 ± 0.166 ^a^	0.661 ± 0.024 ^a^	7.893 ± 0.449 ^a^	8.079 ± 0.631 ^a^	0.412 ± 0.018 ^a^	**96**
**0.75**	26.629 ± 0.915 ^b^	1.497 ± 0.334 ^b^	0.788 ± 0.015 ^b^	20.892 ± 0.813 ^c^	29.100 ± 2.340 ^d^	0.597 ± 0.027 ^c^	**0**
	30.731 ± 1.063 ^c^	0.959 ± 0.156 ^a^	0.704 ± 0.124 ^ab^	20.789 ± 1.662 ^c^	18.530 ± 1.741 ^c^	0.479 ± 0.068 ^b^	**24**
	25.643 ± 1.058 ^b^	0.883 ± 0.065 ^a^	0.609 ± 0.027 ^a^	15.897 ± 0.973 ^b^	13.230 ± 0.528 ^b^	0.406 ± 0.029 ^a^	**48**
	15.285 ± 0.790 ^a^	0.984 ± 0.008 ^a^	0.696 ± 0.010 ^ab^	10.583 ± 0.607 ^a^	10.390 ± 0.643 ^a^	0.458 ± 0.011 ^ab^	**72**
	15.525 ± 0.819 ^a^	0.982 ± 0.009 ^a^	0.670 ± 0.005 ^a^	10.462 ± 0.411 ^a^	10.277 ± 0.397 ^a^	0.431 ± 0.004 ^ab^	**96**
**1**	23.786 ± 0.856 ^c^	0.997 ± 0.018 ^a^	0.726 ± 0.023 ^a^	17.384 ± 0.653 ^d^	17.217 ± 0.614 ^b^	0.533 ± 0.034 ^b^	**0**
	18.275 ± 1.658 ^b^	1.223 ± 0.399 ^a^	0.723 ± 0.062 ^a^	13.058 ± 0.993 ^c^	15.409 ± 0.723 ^b^	0.528 ± 0.049 ^b^	**24**
	35.652 ± 0.964 ^d^	0.940 ± 0.045 ^a^	0.625 ± 0.040 ^a^	16.437 ± 0.720 ^d^	15.623 ± 1.174 ^b^	0.422 ± 0.042 ^a^	**48**
	17.824 ± 1.044 ^b^	0.905 ± 0.088 ^a^	0.484 ± 0.329 ^a^	9.280 ± 1.121 ^b^	8.699 ± 1.190 ^a^	0.432 ± 0.031 ^a^	**72**
	10.597 ± 0.630 ^a^	1.311 ± 0.575 ^a^	0.698 ± 0.019 ^a^	7.456 ± 0.412 ^a^	8.191 ± 1.041 ^a^	0.433 ± 0.043 ^a^	**96**
	**7. Region**						
**0**	41.474 ± 1.885 ^d^	1.557 ± 0.519 ^b^	0.724 ± 0.073 ^b^	31.715 ± 0.599 ^d^	45.698 ± 2.644 ^d^	0.589 ± 0.080 ^c^	**0**
	41.990 ± 3.933 ^d^	0.965 ± 0.037 ^a^	0.684 ± 0.036 ^ab^	28.051 ± 2.441 ^c^	27.600 ± 2.630 ^c^	0.522 ± 0.032 ^bc^	**24**
	35.763 ± 1.846 ^c^	0.947 ± 0.056 ^a^	0.661 ± 0.047 ^ab^	22.919 ± 0.779 ^b^	21.171 ± 1.179 ^b^	0.487 ± 0.063 ^abc^	**48**
	23.420 ± 1.446 ^b^	0.945 ± 0.068 ^a^	0.627 ± 0.032 ^ab^	14.305 ± 0.900 ^a^	13.772 ± 0.615 ^a^	0.411 ± 0.026 ^ab^	**72**
	17.802 ± 0.868 ^a^	1.113 ± 0.309 ^ab^	0.597 ± 0.101 ^a^	11.961 ± 0.825 ^a^	12.681 ± 1.393 ^a^	0.382 ± 0.093 ^a^	**96**
**0.5**	16.158 ± 0.748 ^b^	1.650 ± 0.526 ^b^	0.795 ± 0.045 ^b^	6.322 ± 1.057 ^a^	11.617 ± 0.961 ^b^	0.642 ± 0.036 ^b^	**0**
	34.062 ± 2.017 ^d^	0.931 ± 0.089 ^a^	0.630 ± 0.085 ^a^	24.375 ± 1.626 ^e^	23.777 ± 2.063 ^d^	0.456 ± 0.072 ^a^	**24**
	33.502 ± 3.215 ^d^	0.906 ± 0.076 ^a^	0.634 ± 0.089 ^a^	22.159 ± 0.822 ^d^	22.224 ± 1.542 ^d^	0.435 ± 0.090 ^a^	**48**
	25.024 ± 1.045 ^c^	0.927 ± 0.078 ^a^	0.643 ± 0.022 ^ab^	15.803 ± 0.562 ^c^	14.733 ± 0.649 ^c^	0.419 ± 0.010 ^a^	**72**
	8.097 ± 1.426 ^a^	0.964 ± 0.050 ^a^	0.618 ± 0.049 ^a^	9.387 ± 0.765 ^b^	9.047 ± 0.798 ^a^	0.389 ± 0.034 ^a^	**96**
**0.75**	34.207 ± 2.849 ^d^	1.204 ± 0.344 ^a^	0.769 ± 0.012 ^a^	26.122 ± 2.020 ^d^	31.324 ± 2.750 ^c^	0.615 ± 0.015 ^b^	**0**
	29.716 ± 2.062 ^c^	1.023 ± 0.409 ^a^	0.649 ± 0.138 ^a^	17.183 ± 0.720 ^c^	16.424 ± 0.925 ^b^	0.452 ± 0.120 ^a^	**24**
	28.535 ± 0.524 ^c^	0.850 ± 0.052 ^a^	0.551 ± 0.078 ^a^	17.324 ± 1.109 ^c^	14.457 ± 0.893 ^b^	0.360 ± 0.057 ^a^	**48**
	21.287 ± 1.878 ^b^	0.963 ± 0.013 ^a^	0.658 ± 0.023 ^a^	14.642 ± 0.541 ^b^	14.032 ± 0.536 ^b^	0.443 ± 0.022 ^a^	**72**
	16.691 ± 0.679 ^a^	0.966 ± 0.010 ^a^	0.661 ± 0.025 ^a^	9.456 ± 1.062 ^a^	9.162 ± 0.983 ^a^	0.419 ± 0.022 ^a^	**96**
**1**	26.858 ± 3.199 ^c^	1.036 ± 0.039 ^a^	0.717 ± 0.042 ^b^	18.582 ± 1.965 ^c^	19.402 ± 2.250 ^d^	0.520 ± 0.032 ^a^	**0**
	42.222 ± 1.697 ^d^	0.962 ± 0.028 ^a^	0.669 ± 0.030 ^b^	15.836 ± 0.529 ^b^	15.410 ± 0.660 ^c^	0.454 ± 0.062 ^ab^	**24**
	22.724 ± 1.385 ^b^	0.873 ± 0.051 ^a^	0.560 ± 0.026 ^a^	19.872 ± 0.981 ^c^	17.409 ± 0.870 ^cd^	0.380 ± 0.023 ^a^	**48**
	25.411 ± 0.982 ^bc^	0.874 ± 0.096 ^a^	0.643 ± 0.076 ^ab^	16.116 ± 0.588 ^b^	12.875 ± 0.759 ^b^	0.442 ± 0.075 ^ab^	**72**
	11.522 ± 1.169 ^a^	1.303 ± 0.587 ^a^	0.685 ± 0.073 ^b^	7.197 ± 0.905 ^a^	9.555 ± 0.552 ^a^	0.435 ± 0.069 ^ab^	**96**

a,b,c,d,e: The difference between the means with different letters in the same column is significant (*p* < 0.05).

## Data Availability

The data generated and analyzed during this study are included in this article.

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
