# Peer review of "The Effect of Sodium Bicarbonate Injection on the Physico-Chemical Quality of Post-Harvest Trout"

_foods, 2023, doi:10.3390/foods12132437_

Round 1

Reviewer 1 Report (Previous Reviewer 2)

The authors have satisfactorily addressed most of my concerns. However, the authors should  adjust the size of the figure in the article, such as the Figure 1, so as to improve the Typesetting.

Author Response

Figure 1 size adjusted in the article, so as to improve the Typesetting. Thank you for your interest.

Reviewer 2 Report (Previous Reviewer 4)

No further comments.

Minor editing of English language required.

Author Response

English editing of the paper was edited by the MDPI language editing services. The editing certificate was attached. Thank you for your interest.

This manuscript is a resubmission of an earlier submission. The following is a list of the peer review reports and author responses from that submission.

Round 1

Reviewer 1 Report

The objectives of this manuscript “The effect of sodium bicarbonate injection on the physicochemical quality of post-harvest trout” were to investigate the effectiveness of sodium bicarbonate injections in preventing tissue softening caused by low pH after death in trout. The manuscript is easy to read and understand, however, there are some several scientific concerns about this manuscript.

1.      Lines 134-136, There is something wrong with the format of the formula. Please check it carefully.

2.      Line 145, “U.K” should be revised to “UK”.

3.      Section 3.1, Why wasn't the nutrition composition of rainbow trout measured at different times (24, 48, 72, 96 h) and different regions?

4.      Lines 247-248, “Increased muscle pH at 0.75 and 1 M SBC concentrations resulted in prolonged entry and exit times into rigor”. However, it can only be seen from Figure 1 that the rigor index of 0.75 and 1M SBC group were larger than 0M, there was little significant difference in the storage time of transitional changes between the four groups. Does that contradict the idea of prolonged entry and exit times?

5.      Lines 439-440, the highest or lowest a* value should be a definite value, not a range.

6.      Line 460, “Our results showed” should be revised by “The/this results showed”. It is best not to include personal words in the conclusion.

Try not to use "I, we" and other emotive words.

Author Response

Reviewer 1:

            The objectives of this manuscript “The effect of sodium bicarbonate injection on the physicochemical quality of post-harvest trout” were to investigate the effectiveness of sodium bicarbonate injections in preventing tissue softening caused by low pH after death in trout. The manuscript is easy to read and understand, however, there are some several scientific concerns about this manuscript.

            We would like to thank you for your suggestions and comments. All changes have been added to the revised MS to improve the quality of the MS. Please see them in the text.

            Lines 134-136, There is something wrong with the format of the formula. Please check it carefully.

I apologize for the mistake. The format of the formula was checked according to the journal format and all mistakes were fixed.

Line 145, “U.K” should be revised to “UK”.

It was corrected.

Section 3.1, Why wasn't the nutrition composition of rainbow trout measured at different times (24, 48, 72, 96 h) and different regions?

Since the main purpose of this study is not to observe the change of proximate composition of fish over time, it is to analyze and present the nutritional content of the product before SBC application. Can be removed from operation if desired.

            Lines 247-248, “Increased muscle pH at 0.75 and 1 M SBC concentrations resulted in prolonged entry and exit times into rigor”. However, it can only be seen from Figure 1 that the rigor index of 0.75 and 1M SBC group were larger than 0M, there was little significant difference in the storage time of transitional changes between the four groups. Does that contradict the idea of prolonged entry and exit times?

            The sentence was revised asIn the SBC application groups, excluding the control group, the increased muscle pH, depending on the concentration used, caused the entry and exit times to rigor to be prolonged.’

            Lines 439-440, the highest or lowest a* value should be a definite value, not a range.

            It was corrected as suggested.

            Line 460, “Our results showed” should be revised by “The/this results showed”. It is best not to include personal words in the conclusion.

            They were corrected through the MS.

Reviewer 2 Report

The manuscript “The effect of sodium bicarbonate injection on the physico- 2 chemical quality of post-harvest trout” focuses on investigating the effects of sodium bicarbonate (SBC) injections in preventing tissue softening caused by low pH after death in trout. Four different concentrations of SBC 92 (0,0.50,0.75 and 1M) were injected to prevent protein denaturation and texture deterioration caused by low pH in postmortem fish.

However, there are numerous issues which the authors should address before the manuscript is further considered for publication.

Line 92-93: Why did the authors choose these four concentrations?

Line 187-189: The proliferation of spoilage bacteria in fish meat is also an important factor leading to the change of pH in fish meat. How to eliminate the influence of spoilage bacteria in the change of pH in this result?

Line 238: The expression is not accurate. At present, the conclusion can only prove the change of pH, but "reduces chemical and textural quality losses" has not yet proved it.

Line 292: In the material and method, the fish were divided into 5 equal regions on both sides of their bodies, while in texture analysis, the selected 7 regions do not correspond to the previous 5 regions, which can not fully prove the reason for the texture change of fish at SBC concentration.

It is also necessary to increase sensory indicators to explain the effect of SBC intramuscular injection on fish itself ?

Does this approach have practical significance?

The language needs further modification.

Author Response

Reviewer 2:

            Line 92-93: Why did the authors choose these four concentrations?

            The concentrations determined in our literature study were studied. It was added to the text.

            Line 187-189: The proliferation of spoilage bacteria in fish meat is also an important factor leading to the change of pH in fish meat. How to eliminate the influence of spoilage bacteria in the change of pH in this result?

Bacterial activities can be reduced by using antimicrobial agents. However, since the subject of our study was SBC and its antimicrobial effect was not considered, more physicochemical quality analyzes were focused on.

            Line 238: The expression is not accurate. At present, the conclusion can only prove the change of pH, but "reduces chemical and textural quality losses" has not yet proved it.

The expression was modified. Please check line 238.

            Line 292: In the material and method, the fish were divided into 5 equal regions on both sides of their bodies, while in texture analysis, the selected 7 regions do not correspond to the previous 5 regions, which can not fully prove the reason for the texture change of fish at SBC concentration.

According to the literature study, the regions where injection and texture measurement were made were determined and references were added to the text. These locations were chosen because muscle cell sizes and arrangement vary throughout the body, and textural characteristics differ in different parts of the body.

            It is also necessary to increase sensory indicators to explain the effect of SBC intramuscular injection on fish itself ?

In general, we tried to be in harmony with the parameters in similar studies in the literature, but this situation will be taken into account in our further studies.

            Does this approach have practical significance?

The use of sodium bicarbonate, which is a direct agent on pH, can be used easily in all studies where acid-base balance is important. Convenience can be used in sodium bicarbonate processing technology studies, which are approved by the FDA for use as a food additive. The immersion method may be easier in terms of applicability in enterprises that process high volumes of products.

Reviewer 3 Report

The manuscript provides important results and conclusions regarding the effect of sodium bicarbonate injection on the physicochemical quality of post-harvest trout. The quality of the images, the depth of analysis, and the language of the article are still some way from being published as an academic paper. The specific comments are as follows:

Formatting and language of the manuscript needs further improvement.

Highlights

I think that the current Highlights did not provide exact highlights of this manuscript, number 2-4 especially, the authors are expected to revise these points.

Abstract

The repetition of the highest pH in the groups injected with 0.75M SBC during storage in Line16-17 and Line21.

Introduction

Line31-32 “Several studies” refers to several references, but only one reference was listed in there. Same as the next sentence.

Line 36-38. There are logical problems in this passage.

Line49-51. There are confused sentence.

The author needs to revise the content after reorganizing the logic of the introduction.

Material and Methods

Line 89 to 92. The author wrote three sentences about the information of raw fish, and the information of SBC was interspersed in the three sentences. It is suggested that the author should adjust the content or integrate the three sentences into 1-2 sentences.

Line 127, 134, 136, 163, 164. The format of the equation needs to be standardized.

Line 130, 131. The full name of WHC is not given.

Line 147 to 148. What is the temperature of the refrigerator?

Results and discussion

Line 188 to 189. Missing p-value.

Line 194 to 196. The description of the significance of the samples regarding 1M SBC treatment at 0 h was inconsistent before and after.

Line 198 to 203. There is too much unnecessary repetitive analysis.

Line 208 to 213. The author simply cites the phenomenon without giving the reasons for such changes.

Line 246 to 247. “Several studies” refers to several references, but only one reference was listed in there.

Line 254. Figure1 needs to be redrawn.

Line 264 to 265. Missing p-value.

Line 270 to 279. An in-depth analysis of the reasons for the increase in WHC is needed.

Regarding the results and discussion, the authors simply cite the data and do not provide an in-depth analysis of the reasons for the changes in the data, and the joint analysis among the indicators is missing.

Unfortunately, my decision about this work is rejection.

The language must be improved.

Author Response

Reviewer 3:

            The manuscript provides important results and conclusions regarding the effect of sodium bicarbonate injection on the physicochemical quality of post-harvest trout. The quality of the images, the depth of analysis, and the language of the article are still some way from being published as an academic paper. The specific comments are as follows:

            Thank you for your suggestions and comments. All changes have been added to the revised MS to improve the quality of MS. We hope all the changes we've made will help you change your mind about the release of the revised version of MS.

Formatting and language of the manuscript needs further improvement.

The MS was improved in terms of formatting and language. The editing certificate was also added at the end of the page.

Highlights

I think that the current Highlights did not provide exact highlights of this manuscript, number 2-4 especially, the authors are expected to revise these points.

The highlights were revised.

Abstract

The repetition of the highest pH in the groups injected with 0.75M SBC during storage in Line16-17 and Line21.

It was corrected.

Introduction

Line31-32 “Several studies” refers to several references, but only one reference was listed in there. Same as the next sentence.

References were added to the text.

Line 36-38. There are logical problems in this passage.

The sentence was revised.

Line49-51. There are confused sentence.

The author needs to revise the content after reorganizing the logic of the introduction.

The sentence was modified.

Material and Methods

Line 89 to 92. The author wrote three sentences about the information of raw fish, and the information of SBC was interspersed in the three sentences. It is suggested that the author should adjust the content or integrate the three sentences into 1-2 sentences.

The paragraph was modified.

Line 127, 134, 136, 163, 164. The format of the equation needs to be standardized.

The format of the formula was checked according to the journal format and all mistakes were fixed.

Line 130, 131. The full name of WHC is not given.

It was corrected as suggested.

Line 147 to 148. What is the temperature of the refrigerator?

The temperature of the refrigerator was added to the text.
Results and discussion

Line 188 to 189. Missing p-value.

It was added to the text.

Line 194 to 196. The description of the significance of the samples regarding 1M SBC treatment at 0 h was inconsistent before and after.

It was rewritten.

Line 198 to 203. There is too much unnecessary repetitive analysis.

It was rewritten.

Line 208 to 213. The author simply cites the phenomenon without giving the reasons for such changes.

The paragraph was rewritten.

Line 246 to 247. “Several studies” refers to several references, but only one reference was listed in there.

References were added to the text.

Line 254. Figure1 needs to be redrawn.

It was corrected.

Line 264 to 265. Missing p-value.

It was added to the text.

Line 270 to 279. An in-depth analysis of the reasons for the increase in WHC is needed.

The paragraph was modified.

Regarding the results and discussion, the authors simply cite the data and do not provide an in-depth analysis of the reasons for the changes in the data, and the joint analysis among the indicators is missing.

Necessary changes were added in the MS.

Reviewer 4 Report

This study aimed to determine the effectiveness of sodium bicarbonate (SBC) injections in preventing tissue softening caused by low pH after death in trout. Injection of different molar concentrations of SBC in rainbow trout (Oncorhynchus mykiss; 0M, 0.5M, 0.75M, and 1M) after harvest was conducted and the product quality was assessed at 0, 24, 48, 72, and 96 hours of ice storage.

This manuscript presents an interesting subject, and although similar studies have been conducted on this focus, I can notice novelty in this work.

However, there are still some spots should be improved as I recommend a major revision before acceptance for publication. Some information in the manuscript is missing and sometimes appear confusing sentences.

There are two publications on this focus and species which the Author did not consider. They can represent a useful background and to compare with. Therefore, I strongly suggest the authors to reformulate the text throughout the manuscript.

The following are my remarks in detail

Abstract

Very satisfactory

Page 1- Lines 8-9 Replace with Post-mortem decrease in the pH of fish tissue.......

Page 1- Line 9 You should correct this sentence “We hypothesized that maintaining a high pH of the fish tissue high after death would prevent…..”

Introduction

This section should be improved!

Page 1- Lines 26-38 This first part of the introduction is badly written, with poorly connected and poorly flowing sentences; however this covers more or less the entire introduction. Also, as far as aim is it pre-harvest or post-harvest? Sentences confuse the reader, more clarity is needed.

Materials and Methods

Page 2- Line 92 add the standard deviation to the length and weight

Page 2- Line 93 …..as the control group

Page 2- Line 95  5 equal regions on both sides of their bodies (both lateral sites of fish)’ specify better

Page 3- Line 107  Change with Protein contents of samples were determined by the Kjeldahl                     procedure

Page 3- Line 110  Delete In addition 

Results and Discussion

A lot of data piled up without getting to the point. Do not include a lot of specific data in the description of results, all the specific values have been presented in the tables.

Please check this text section carefully, there are many wrong data

Page 5- Line 195 ‘The 0.75 M and 1 M SBC treatment groups had significantly higher pH values at       0h (p<0.05)’.  It is not true for 1 M SBC, as can be seen from the table.

Page 5- Line 198.  The 0.75 M and 1 M groups had the highest pH values (p<0.05)’, It is not true,

replace with ‘The 0.75 M and 1 M groups had the highest pH values, although not significant’.

Page 5- Line 198.  Delete this sentence The 0.75 M SBC group had the highest pH value 200 compared to the control group (pH = 7.051; p<0.05). It's not needed.

Page 5- Line 207.  The lowest and highest pH values in the 0 M injection group were 6.549 at 72 h and 6.789 at 0 h. This sentence has already been above written.

Page 6- Lines 225-226  In the 0.75 M and 1 M injection groups, the highest pH was detected in the fourth region (p<0.05). It is no true for 0.75 M, since the fourth region did not differ from the third and five regions.

Page 6- Lines 225-226 The 0 M group had the lowest, and the 0.75 M group had the highest pH in the second, third, fourth, and fifth regions.  Delete this sentence, confuses the reader.

Page 7- Lines 263-265 Table 4 summarizes the changes in WHC in trout muscles. Specifically, we observed an increase in WHC at 0.75% and 1% SBC levels. WHC did not differ significantly with storage duration. The 1 M and 0.75 M groups had the highest WHC at 48 h (94.967% and 94.812%, respectively). Check the letters on the same line and on the same column.

Page 8- Lines 302-304 …………the lowest hardness of 6.224 N was noted in the 0 M group at 24h.

As can be seen from the table the lowest hardness of 6.224 N was noted in the 0 M group at 96 hours.

Page 8- Lines 306-307  The highest and the lowest cohesiveness values were 0.819% and 0.678% at 96 h in the 0 M group. This sentence is not correct, please rewrite….. The highest and lowest cohesion values were 0.819% and 0.678% at day 0 and 96 hours, respectively, in the 0 M group.

Page 8- Lines 307-309  The highest and lowest gummy values for the 1 M group were 12,871 N at 24 hours and 4,779 N at 48 hours. Again this doesn't seem correct, please rewrite with the right information.

Page 13- Lines 324-326  ….. highest value and 0.976% at the lowest 0.75M concentration in 48 hours of storage. Write this sentence more clearly, confuses the reader.

Page 13- Lines 346-347   The highest chewiness value of 22,028 N was recorded in the 0.5 M group at day 0, and the lowest value of 7.489 N was recorded in the 0.5 M group at 96 h…….

            Replace with: Both the highest (22,028 N) and lowest (7.489 N) chewiness values were recorded in the 0.5 M group, at 0 and at 96 h, respectively. Likewise the highest and lowest resilience values.….

Page 13- Lines 355-356.….and the lowest value of 8,115 N was recorded in the 0.5 M group at 96 h. In the table in the 0.75 M group at 96 h appears this value 0.963±0.047. It is a mistake? Please check.

Page 13- Lines 359-360…..and the lowest value of 0.665% was recorded in the 0.5 M group at 24 h. The correct value is 0.628. Please check and correct.

Page 13- Lines 363-364…..The highest and lowest resilience values of 0.668% and 0.398% were recorded in the 0 M group at day 0 and 96 h, respectively. Please check and correct the values.

Page 13- Lines 371-372…..The highest and lowest springiness values of 1.726% and 0.920% were recorded in the 0.5 M group at 0 h and 48 h, respectively.

The correct value of the lowest springiness is 0.918 in the 0.75 M group at 24 h. Please check and correct.

Page 14- Line 387…..and the lowest value of 12,122 N was observed in the 0.5 M group at 96 h.

Please check and correct. Both highest and lowest hardness were in the 1 M group. The correct value of the lowest hardness is 10.597 in the 1 M group at 96 h.

Page 14- Lines 389-390  The highest cohesiveness value was 0.811%, as noted in the 0.5 M group at 0 h,…. 0.811% is also the cohesiveness value observed in the 0 group at 0 h. Please check and correct.

Page 14- Line 403  The highest springiness value noted was 1.557% in the 0 M group at 0 h…..

The correct value of the highest springiness was 1.650% in the 0.5 M group at 0 h. Please check and correct.

These are the publications on this focus  that are missing

R. El Rammouz, J. Abboud, M. Abboud, A. El Mur, S. Yammine and B. Jammal 2013 pH, Rigor Mortis and Physical Properties of Fillet in Fresh Water Fish: The Case of Rainbow Trout (Oncorynchus mykiss). Journal of Applied Sciences Research, 9(11): 5746-5755.

Elif Tugçe Aksun,  Bahar Karakaya Tokur,  2014. Effects of sodium bicarbonate injection on sensory and chemical qualities of rainbow trout during iced storage. Ege J Fish Aqua Sci 31(2): 97-104. DOI: 10.12714/egejfas.2014.31.2.06

English should be improved and made more fluent

Author Response

Reviewer 4:

            This study aimed to determine the effectiveness of sodium bicarbonate (SBC) injections in preventing tissue softening caused by low pH after death in trout. Injection of different molar concentrations of SBC in rainbow trout (Oncorhynchus mykiss; 0M, 0.5M, 0.75M, and 1M) after harvest was conducted and the product quality was assessed at 0, 24, 48, 72, and 96 hours of ice storage.

This manuscript presents an interesting subject, and although similar studies have been conducted on this focus, I can notice novelty in this work.

However, there are still some spots should be improved as I recommend a major revision before acceptance for publication. Some information in the manuscript is missing and sometimes appear confusing sentences.

There are two publications on this focus and species which the Author did not consider. They can represent a useful background and to compare with. Therefore, I strongly suggest the authors to reformulate the text throughout the manuscript.

The following are my remarks in detail

            We would like to thank you for your suggestions and comments. All changes have been added to the revised MS to improve the quality of the MS. Please see them in the text.

Abstract

Page 1- Lines 8-9 Replace with Post-mortem decrease in the pH of fish tissue.......

It was corrected as suggested.

Page 1- Line 9 You should correct this sentence “We hypothesized that maintaining a high pH of the fish tissue high after death would prevent…..”

It was corrected as suggested.

Introduction

Page 1- Lines 26-38 This first part of the introduction is badly written, with poorly connected and poorly flowing sentences; however this covers more or less the entire introduction. Also, as far as aim is it pre-harvest or post-harvest? Sentences confuse the reader, more clarity is needed.

The paragraph was modified.

Material and Methods

Page 2- Line 92 add the standard deviation to the length and weight

I apologize for the mistake we missed. It was added to the text.

Page 2- Line 93 …..as the control group

It was corrected as suggested.

Page 2- Line 95  ‘5 equal regions on both sides of their bodies (both lateral sites of fish)’ specify better

It was corrected as suggested.

Page 3- Line 107  Change with Protein contents of samples were determined by the Kjeldahl procedure

It was corrected as suggested.

Page 3- Line 110  Delete In addition 

It was corrected as suggested.

Results and discussion

Page 5- Line 195 ‘The 0.75 M and 1 M SBC treatment groups had significantly higher pH values at       0h (p<0.05)’.  It is not true for 1 M SBC, as can be seen from the table.

I apologize for the mistake. It was rewritten.

Page 5- Line 198.  ‘The 0.75 M and 1 M groups had the highest pH values (p<0.05)’, It is not true,

replace with ‘The 0.75 M and 1 M groups had the highest pH values, although not significant’.

 I apologize for the mistake. It was corrected as suggested. It was rewritten.

Page 5- Line 198.  Delete this sentence The 0.75 M SBC group had the highest pH value 200 compared to the control group (pH = 7.051; p<0.05). It's not needed.

It was corrected as suggested.

Page 5- Line 207.  The lowest and highest pH values in the 0 M injection group were 6.549 at 72 h and 6.789 at 0 h. This sentence has already been above written.

 It was corrected as suggested.

Page 6- Lines 225-226  In the 0.75 M and 1 M injection groups, the highest pH was detected in the fourth region (p<0.05). It is no true for 0.75 M, since the fourth region did not differ from the third and five regions.

 I apologize for the mistake. It was rewritten.

Page 6- Lines 225-226 The 0 M group had the lowest, and the 0.75 M group had the highest pH in the second, third, fourth, and fifth regions.  Delete this sentence, confuses the reader.

It was corrected as suggested.

Page 7- Lines 263-265 Table 4 summarizes the changes in WHC in trout muscles. Specifically, we observed an increase in WHC at 0.75% and 1% SBC levels. WHC did not differ significantly with storage duration. The 1 M and 0.75 M groups had the highest WHC at 48 h (94.967% and 94.812%, respectively). Check the letters on the same line and on the same column.

 I apologize for the mistake. It was rewritten.

Page 8- Lines 302-304 …………the lowest hardness of 6.224 N was noted in the 0 M group at 24h.

As can be seen from the table the lowest hardness of 6.224 N was noted in the 0 M group at 96 hours.

  I apologize for the mistake. It was rewritten.

Page 8- Lines 306-307 The highest and the lowest cohesiveness values were 0.819% and 0.678% at 96 h in the 0 M group. This sentence is not correct, please rewrite….. The highest and lowest cohesion values were 0.819% and 0.678% at day 0 and 96 hours, respectively, in the 0 M group.

 I apologize for the mistake. It was rewritten.

Page 8- Lines 307-309  The highest and lowest gummy values for the 1 M group were 12,871 N at 24 hours and 4,779 N at 48 hours. Again this doesn't seem correct, please rewrite with the right information.

 I apologize for the mistake. It was rewritten.

Page 13- Lines 324-326  ….. highest value and 0.976% at the lowest 0.75M concentration in 48 hours of storage. Write this sentence more clearly, confuses the reader.

 It was corrected.

Page 13- Lines 346-347   The highest chewiness value of 22,028 N was recorded in the 0.5 M group at day 0, and the lowest value of 7.489 N was recorded in the 0.5 M group at 96 h…….

            Replace with: Both the highest (22,028 N) and lowest (7.489 N) chewiness values were recorded in the 0.5 M group, at 0 and at 96 h, respectively. Likewise the highest and lowest resilience values.….

It was corrected as suggested. It was rewritten.

Page 13- Lines 355-356.….and the lowest value of 8,115 N was recorded in the 0.5 M group at 96 h. In the table in the 0.75 M group at 96 h appears this value 0.963±0.047. It is a mistake? Please check.

It was corrected.

Page 13- Lines 359-360…..and the lowest value of 0.665% was recorded in the 0.5 M group at 24 h. The correct value is 0.628. Please check and correct.

 I apologize for the mistake. It was corrected in the text.

Page 13- Lines 363-364…..The highest and lowest resilience values of 0.668% and 0.398% were recorded in the 0 M group at day 0 and 96 h, respectively. Please check and correct the values.

 I apologize for the mistake. It was corrected in the text.

Page 13- Lines 371-372…..The highest and lowest springiness values of 1.726% and 0.920% were recorded in the 0.5 M group at 0 h and 48 h, respectively.

The correct value of the lowest springiness is 0.918 in the 0.75 M group at 24 h. Please check and correct.

 I apologize for the mistake. It was corrected in the text.

Page 14- Line 387…..and the lowest value of 12,122 N was observed in the 0.5 M group at 96 h.

Please check and correct. Both highest and lowest hardness were in the 1 M group. The correct value of the lowest hardness is 10.597 in the 1 M group at 96 h.

 I apologize for the mistake. It was corrected in the text.

Page 14- Lines 389-390  The highest cohesiveness value was 0.811%, as noted in the 0.5 M group at 0 h,…. 0.811% is also the cohesiveness value observed in the 0 group at 0 h. Please check and correct.

 I apologize for the mistake. It was corrected in the text.

Page 14- Line 403  The highest springiness value noted was 1.557% in the 0 M group at 0 h…..

The correct value of the highest springiness was 1.650% in the 0.5 M group at 0 h. Please check and correct.

 I apologize for the mistake. It was corrected in the text.

These are the publications on this focus  that are missing

  1. El Rammouz, J. Abboud, M. Abboud, A. El Mur, S. Yammine and B. Jammal 2013 pH, Rigor Mortis and Physical Properties of Fillet in Fresh Water Fish: The Case of Rainbow Trout (Oncorynchus mykiss). Journal of Applied Sciences Research, 9(11): 5746-5755.

Elif Tugçe Aksun,  Bahar Karakaya Tokur,  2014. Effects of sodium bicarbonate injection on sensory and chemical qualities of rainbow trout during iced storage. Ege J Fish Aqua Sci 31(2): 97-104. DOI: 10.12714/egejfas.2014.31.2.06

Related references have been added to the article.